# Lateralizing Characteristics of Morphometric Changes to Hippocampus and Amygdala in Unilateral Temporal Lobe Epilepsy with Hippocampal Sclerosis

**DOI:** 10.3390/medicina58040480

**Published:** 2022-03-26

**Authors:** Hyunjin Jo, Jeongsik Kim, Dongyeop Kim, Yoonha Hwang, Daewon Seo, Seungbong Hong, Young-Min Shon

**Affiliations:** 1Samsung Medical Center, Department of Neurology, Sungkyunkwan University School of Medicine, Seoul 06355, Korea; bell530@naver.com (H.J.); js.brain.kim@samsung.com (J.K.); daewon3.seo@samsung.com (D.S.); seungbong.hong@samsung.com (S.H.); 2Department of Neurology, Seoul Hospital, Ewha Womans University College of Medicine, Seoul 03760, Korea; hap2028@naver.com; 3Department of Neurology, The Catholic University of Korea Eunpyeong St. Mary’s Hospital, Seoul 07345, Korea; yoonhaha10@gmail.com; 4Department of Medical Device Management and Research, Samsung Advanced Institute for Health Sciences & Technology (SAHIST), Sunkyunkwan University, Seoul 06355, Korea

**Keywords:** hippocampal sclerosis, hippocampus, amygdala, subfield analysis, temporal lobe epilepsy

## Abstract

*Background and**Objective:* In the present study, a detailed investigation of substructural volume change in the hippocampus (HC) and amygdala (AMG) was performed and the association with clinical features in patients with mesial temporal lobe epilepsy with hippocampal sclerosis (TLE-HS) determined. *Methods:* The present study included 22 patients with left-sided TLE-HS (LTLE-HS) and 26 patients with right-sided TLE-HS (RTLE-HS). In addition, 28 healthy controls underwent high-resolution T2-weighted image (T2WI) and T1-weighted image (T1WI) MRI scanning. Subfield analysis of HC and AMG was performed using FreeSurfer version 6.0. *Results*: Patients with TLE-HS showed a decrease in the volume of substructures in both HC and AMG, and this change was observed on the contralateral side and the ipsilateral side with HS. The volume reduction pattern of substructures showed laterality-dependent characteristics. Patients with LTLE-HS had smaller volumes of the ipsilateral subiculum (SUB), contralateral SUB, and ipsilateral cortical nucleus of AMG than patients with RTLE-HS. Patients with RTLE-HS had reduced ipsilateral cornu ammonis (CA) 2/3 and contralateral cortico-amygdaloid transition area (CAT) volumes. The relationship between clinical variables and subregions was different based on the lateralization of the seizure focus. Focal to bilateral tonic-clonic seizures (FTBTCS) was associated with contralateral and ipsilateral side subregions only in LTLE-HS. The abdominal FAS was associated with the volume reduction of AMG subregions only in LTLE-HS, but the volume reduction was less than in patients without FAS. *Conclusions:* The results indicate that unilateral TLE-HS is a bilateral disease that shows different laterality-dependent characteristics based on the subfield analysis of HC and AMG. Subfield volumes of HC and AMG were associated with clinical variables, and the more damaged substructures depended on laterality in TLE-HS. These findings support the evidence that LTLE-HS and RTLE-HS are disparate epilepsy entities rather than simply identical syndromes harboring a mesial temporal lesion. In addition, the presence of FAS supports good localization value, and abdominal FAS has a high localization value, especially in patients with LTLE-HS.

## 1. Introduction

Hippocampal sclerosis (HS) is a common pathology observed in mesial temporal lobe epilepsy (TLE) and other epilepsy syndromes. HS is identified in 36.4% of epilepsy surgery specimens [1].

The use of brain magnetic resonance imaging (MRI) for the noninvasive diagnosis of HS should be a mandatory part of the diagnostic workup of patients with TLE. The MRI features of HS include atrophy and/or high signal intensity of the hippocampus (HC) on T2-weighted (T2WI) and fluid-attenuated inversion recovery (FLAIR) images [2,3,4,5,6]. However, hippocampal volume alone is not a reliable predictor of postsurgical outcomes in patients with mesial TLE with HS (TLE-HS) [7,8].

Neuropathological investigations have described that HS has diverse patterns of neuronal cell loss within hippocampal subfields, and various attempts have been made to classify specific patterns of hippocampal neuronal cell loss and correlate subtypes with clinical features and postsurgical outcomes [9]. The International League Against Epilepsy (ILAE) proposed a classification of HS. HS ILAE type 1 shows predominant neuronal cell loss and gliosis in the cornu ammonis (CA)1 and CA4 regions that are more often associated with a history of initial precipitating injuries before the age of 5 years with early seizure onset and favorable postsurgical seizure control [10]. As clinical features and postsurgical outcomes differ by the cell loss pattern of hippocampal subfields, MRI techniques for estimating hippocampal subfield volumes have been developed. This has led to the development of hippocampal subfield techniques that can be applied to clinically acquired (i.e., ≤3T) MRI data [11,12].

In addition to the HC, other structures in the brain are affected by TLE-HS. Extensive sclerosis of adjacent structures in the medial temporal lobe, including the amygdala (AMG) and parahippocampal gyrus (PHG), may also be observed [6,13]. Epilepsy duration is reportedly not limited to HC volume and correlates with AMG volume loss [14], while pure AMG atrophy, without hippocampal atrophy, is associated with age at epilepsy onset and frequency of focal to bilateral tonic-clonic seizures (FTBTCS) [15].

Therefore, evaluation of the subfield volumes of HC and its adjacent structures, especially AMG, may provide useful information for predicting clinical features and postsurgical outcomes in patients with TLE-HS. However, the manual delineation of these subfield volumes is a labor-intensive procedure. Several studies have developed automated methods to reduce workload and increase reproducibility. Here, we used hippocampal subfield volumetry (HSV), which is widely used in open-source automatic segmentation software (FreeSurfer version 6.0). HSV is a useful and adequate replacement for manual tracing that saves time and cost in subfield volume analyses [16,17,18,19].

This study aimed to determine which subfield volume is related to specific clinical features of TLE-HS through automatic subfield volume measurements using HSV. It also attempted to determine whether these associations showed different characteristics according to the lateralization of the seizure focus.

## 2. Methods

### 2.1. Participants

Mesial TLE (MTLE) with HS patients were recruited among patients who were admitted for video-EEG monitoring at our institute between March 2015 and December 2018. The study included 48 MTLE patients with confirmed unilateral HS on brain MRI (19 men; mean ± standard deviation [SD] age, 31.4 ± 11.6 years), including 22 patients with left-sided MTLE-HS (LTLE-HS) and 26 patients with right-sided MTLE-HS (RTLE-HS). HS was defined as hippocampal volume loss and internal structure disruption on T1-weighted images (T1WI) and/or hyperintensities on T2WI and FLAIR images [20] and confirmed by a neuroradiologist (Kim ST) specializing in epilepsy. Patients with the following criteria were excluded: a significant medical history of acute encephalitis, meningitis, severe head trauma, or ischemic encephalopathy; suspicious epileptogenic lesions (e.g., tumor, cortical dysplasia or vascular malformation) on MRI other than ipsilateral HS at the abnormal electroencephalogram side; or epileptic paroxysms in extra-temporal regions on an electroencephalogram.

This study included 28 healthy volunteers as controls (38.4 ± 6.4 years; 12 men, 16 women). The control participants had no abnormal findings on neurological examination or brain MRI.

The medical records of the patients were searched to determine the age at the time of the MRI, age at epilepsy onset, duration of epilepsy, frequency of focal aware seizures (FAS), frequency of focal impaired awareness seizures (FIAS), history of childhood febrile seizures, history of FTBTCS, presence of abdominal/psychic FAS, and surgical outcome. Patients who underwent surgery (standardized amygdalohippocampectomy) were followed up postoperatively for up to 2 years, and their outcomes were assessed using the ILAE outcome classification system (Table 1) [21].

### 2.2. MRI Acquisition

All study participants underwent T1-weighted MRI using a 3-Tesla scanner (Achieva; Philips Medical Systems, Best, The Netherlands) with a 16-channel head coil. The anatomical images were obtained using the same scanning parameters as those in previously published studies [22].

### 2.3. MRI Analysis

For each patient, T1-weighted MRI volume was preprocessed using the standard FreeSurfer recon-all pipeline (version 6.0; https://surfer.nmr.mgh.harvard.edu/) (Accessed on 31 January 2021) for quantitative automated segmentation and cortical parcellation (Appendix A). After the completion of the recon-all process, all cortical and subcortical segmentations were visually inspected, and manual editing was performed to correct the boundaries of the pial and white matter surfaces. Participants who required more than 30 min of correction time were excluded to avoid processing bias. Hippocampal subfield and AMG nuclei volume estimates were subsequently obtained using the development version (freesurfer-linux-centos8) of FreeSurfer (https://surfer.nmr.mgh.harvard.edu/fswiki/HippocampalSubfieldsAndNeucleiOfAmygdala) (Accessed on 31 January 2021). The algorithm and segmentation protocol of this tool referenced the methods described in previously published papers [23].

Thirteen regions were calculated, including CA1, CA2/3, CA4, granule cell layer of the DG (DG), fimbria, subiculum (SUB), presubiculum, parasubiculum, molecular layer, hippocampus-amygdala-transition area (HATA), hippocampal tail, whole HC, and hippocampal fissure. Only CA1 head, CA1, CA2/3 head, CA2/3, CA4 head, CA4, DG head, DG body, SUB head, and SUB body volumes were used to obtain a more concise and clear description.

Seven regions of the AMG were calculated, including the lateral nucleus (La), basal nucleus (Ba), accessory basal (AB), central (Ce), medial (Me), cortical (Co), and cortico-amygdaloid transition area (CAT).

### 2.4. Statistical Analysis

All statistical analyses were performed using the Statistical Package for the Social Sciences (SPSS) for Windows version 21.0 (SPSS Inc., Chicago, IL, USA).

Subfield volumes were corrected for intracranial volume, and statistical tests were corrected for multiple comparisons using the false discovery rate procedure. The volumes of the analyzed structures were compared with those of 28 age- and sex-matched controls and transformed into z-scores. In each subfield, volume z-scores were calculated using the following formula: volume z-score = (volume subject—mean volume)/SD volume.

Age differences between groups were analyzed using one-way analysis of variance, while the sex distribution was analyzed using the chi-square test. The data were tested for normality using the Shapiro–Wilk test. Group comparisons were performed using the independent sample *t*-test (normally distributed data, *p* < 0.05) or unpaired Mann–Whitney *U* test (data not normally distributed, *p* < 0.05). Associations between categorical relationships, including sex, history of childhood febrile seizure, and history of FTBTCS, were investigated using the chi-square test of independence.

Associations between subfield volume and clinical data, including age at epilepsy onset, duration of epilepsy, frequency of FAS, and frequency of FIAS, were investigated using Spearman’s correlation coefficient.

### 2.5. Ethical Approval

Our screening and study procedures were approved and monitored by the Institutional Review Board of Samsung Medical Center (no. 2017-07-112-002). Informed written consent was obtained from all study participants prior to enrollment, and they were financially compensated for their participation.

## 3. Results

### 3.1. Participant Characteristics

In the present study, 22 and 26 patients had LTLE-HS and RTLE-HS, respectively. No significant differences in age or sex distribution were observed between controls and patients with TLE-HS. There were no significant differences in age at epilepsy onset, duration of epilepsy, frequency of FAS and FIAS, history of febrile seizures, history of FTBTCS, or psychic FAS between the LTLE-HS and RTLE-HS patients. Abdominal FAS was observed more frequently in the LTLE-HS group than in the RTLE-HS group (*p =* 0.023).

Among the 22 LTLE-HS patients, 17 underwent surgical resection and had a good postoperative prognosis (ILAE outcome scale class 1). Among the 26 RTLE-HS patients, 20 underwent surgical resection; the postoperative outcome was ILAE class 2 for only one patient versus ILAE class 1 for the remaining 19 patients. The detailed demographic information is presented in Table 1.

### 3.2. Volumetric Analyses of HC and AMG Subfields

Overall, HC and AMG mean subfield volumes were smaller in patients than in controls (negative z-scores). In both groups, the mean z-scores of the subfield volumes on the ipsilateral side were smaller than those on the contralateral side (Appendix A).

Patients with LTLE-HS had significantly reduced volumes of the ipsilateral SUB head (*p =* 0.048), contralateral SUB head (*p =* 0.048), and ipsilateral Co (*p =* 0.002) compared with patients with RTLE-HS. Furthermore, RTLE-HS patients had significantly reduced volumes of the ipsilateral CA2/3 head (*p =* 0.047) and contralateral CAT (*p =* 0.013) compared to LTLE-HS patients (Appendix A, Figure 1).

### 3.3. Hippocampal Subfield Volumes and Clinical Correlations

In both patients with LTLE-HS and RTLE-HS, the duration of epilepsy was significantly correlated with hippocampal subfield volumes. In the LTLE-HS group, the volumes of the ipsilateral CA1 head (rho = −0.499, *p =* 0.045), CA1 (rho = −0.484, *p =* 0.045), CA4 head (rho = −0.600, *p =* 0.019), and DG head (rho = −0.587, *p =* 0.008) were negatively correlated with epilepsy duration. In RTLE-HS, all subfield volumes on the ipsilateral side were negatively correlated with epilepsy duration (*p <* 0.05; Table 2).

A history of previous FTBTCS was associated with ipsilateral CA1, DG, and contralateral DG head volume reduction in LTLE-HS (*p =* 0.044, *p =* 0.048, and *p =* 0.009, respectively; Figure 2). However, in RTLE-HS, no subregion showed a significant volume reduction associated with a history of generalized seizures.

No significant relationship was observed between age at epilepsy onset, frequency of FAS, frequency of FIAS, history of febrile seizures, presence of abdominal/psychic FAS, and hippocampal subfield volumes.

### 3.4. AMG Subfield Volumes and Clinical Correlations

In LTLE-HS, onset age was positively correlated with ipsilateral Ce and Me volumes (rho = 0.442, *p =* 0.039 and rho = 0.483, *p =* 0.023, respectively), and the duration of epilepsy was negatively correlated with the volumes of the ipsilateral Ce (rho = −0.535, *p =* 0.010) and Me (rho = −0.535, *p =* 0.010). The subfield volumes of AMG and the frequencies of FAS and FIAS were not correlated in patients with LTLE-HS.

In RTLE-HS, the duration of epilepsy was negatively correlated with the volumes of the ipsilateral Me (rho = −0.392, *p =* 0.048) and Co (rho = −0.407, *p =* 0.039), and the frequency of FAS was positively correlated with the volume of the ipsilateral Me (rho = 0.476, *p =* 0.014; Table 2). Age at epilepsy onset and frequency of FIAS were not associated with AMG subfield volumes in patients with RTLE-HS.

In both the LTLE-HS and RTLE-HS patients, no correlation was observed between a history of febrile seizure and subfield AMG volume.

Patients with a history of FTBTCS in LTLE-HS had lower ipsilateral La, Ba, Ce, CAT, and contralateral La volumes (*p =* 0.005, *p* = 0.005, *p =* 0.030, *p =* 0.040, and *p =* 0.040, respectively) than patients without such a history (Figure 2). However, the history of generalized seizures in RTLE-HS did not correlate with the subfield AMG volume.

Intriguingly, in the LTLE-HS group, smaller ipsilateral AB and CAT volumes (*p =* 0.035 and *p =* 0.035, respectively) were observed in patients without versus those with abdominal FAS (Figure 2). However, psychic FAS was not correlated with the subfield volume of the AMG in patients with LTLE-HS. In the RTLE-HS group, neither abdominal FAS nor psychic FAS were significantly correlated with the subfield AMG volume.

## 4. Discussion

In the present study, subfield volumes on MRI were automatically measured and compared with the clinical features of patients with TLE-HS. Notably, significant volume loss was observed on the contralateral and ipsilateral sides of the lesion compared with the controls, even if the ipsilateral side was significantly smaller than the contralateral side. Our results, similar to those of previous studies [24,25,26,27], support the hypothesis that unilateral TLE-HS can be considered a bilateral disease and that damage is not restricted to the side of seizure onset.

### 4.1. Inhomogeneity of LTLE-HS and RTLE-HS Volume Losses

When comparing the LTLE-HS and RTLE-HS, a more pronounced volume reduction was observed in the different subregions. The volumes of the ipsilateral SUB, contralateral SUB, and ipsilateral Co were significantly smaller in LTLE-HS patients than those in RTLE-HS patients. The volumes of the ipsilateral CA2/3 head and contralateral CAT were smaller in the RTLE-HS group than in the LTLE-HS group. These results indicated that subfield alterations in particular regions depend on the seizure focus lateralization.

TLE laterality has been reported in many studies; however, the results have been inconsistent. In several studies, although bilateral changes in LTLE [28,29] or RTLE [30,31] increased, in other studies, the changes were equal [32]. However, there are few studies on the laterality of subfield alteration. Keller et al. reported that the volumes of the contralateral presubiculum and HATA in LTLE were smaller than those in RTLE and that the contralateral hippocampal tail was smaller in RTLE [19].

In the present study, we confirmed that subfield alteration differs by laterality; this difference is not limited to HC, and it also applies to AMG. Therefore, subfield analysis of AMG, as well as HC, in TLE-HS patients should be considered.

Depending on laterality, the relevant clinical variables were slightly different, as were the subregions associated with each clinical variable.

### 4.2. Inhomogeneous Clinical and MRI Features between LTLE-HS and RTLE-HS

Various clinical factors have been associated with atrophy severity in HC and AMG. This correlation is influenced by laterality, indicating that LTLE-HS and RTLE-HS may correspond to different types of epilepsy.

Previous studies reported different results regarding the association between hippocampal atrophy and age at epilepsy onset (relevant [33,34] or not [14,19,35]), duration of epilepsy (relevant [14,33,34,35] or not [19,36]), frequency of FIAS (relevant [27,37] or not [19,20,36,37,38]), presence of febrile seizure (relevant [14,33,35,36] or not [19]), and a higher number of lifetime FTBTCS (relevant [39,40,41] or not [14,19,36]). Our results show that the duration of epilepsy and history of FTBTCS may be associated with hippocampal subfield atrophy, indicating that hippocampal subfield volume loss correlates with disease chronicity and severity.

The relationship between amygdalar subregion volume and clinical features has been reported in only a few studies. The degree of atrophy in extrahippocampal structures is associated with the degree of hippocampal atrophy [42]; however, whether the degree of amygdalar atrophy correlates with clinical features has yet to be fully investigated. Duration of epilepsy [14,27], history of febrile seizure [36], and FTBTCS [43] are reportedly associated with amygdalar atrophy. However, the relationship between amygdalar atrophy and the duration of epilepsy, febrile seizure, frequency of FIAS, and FTBTCS was determined in only a few studies. [14,36,42].

In the present study, the AMG subfield volume was associated with age at epilepsy onset, duration of epilepsy, frequency of FAS, presence of abdominal FAS, and history of FTBTCS. This result shows that in TLE-HS, AMG and HC are associated with disease chronicity and severity.

In addition, even if the clinical features were very similar, volume loss was observed in different subregions depending on the side of the ictal onset. For example, disease duration corresponded to volume loss only in the CA1, CA4 head, and DG head in LTLE-HS patients versus diffuse hippocampal atrophy in RTLE-HS patients. This discrepancy provides supportive evidence that LTLE-HS and RTLE-HS are disparate epilepsy entities rather than simply identical syndromes harboring a mesial temporal lesion. Avery et al. reported that the degree of correlation between the asymmetry index and age at onset is strongly influenced by the side of seizure onset based on single-photon emission computerized tomography [44]. However, the association between laterality and clinical variables has been investigated in only a few subfields analysis studies. The results of the present study can aid in the elucidation of the lateralizing characteristics of morphometric changes in the mesial temporal regions in unilateral TLE-HS.

### 4.3. Bilateral and Widespread Damage from FTBTCS in LTLE-HS

A history of FTBTCS was associated with subfield HC and AMG volume losses only in LTLE-HS and was associated with volume loss of the contralateral and ipsilateral sides.

The presence of FTBTCS has been associated with more extensive HS [41], multifocal irritative areas [45], and/or more extended hypometabolism based on positron emission tomography (PET) [46].

These results indicate that the presence of FTBTCS reflects a more active and extensive network of epileptogenicity (possibly involving the contralateral side) in TLE. Therefore, FTBTCS appears to affect the other bilateral sides compared with other clinical variables.

In previous studies, multiple factors that increase the vulnerability of the left hemisphere have been suggested [47]; thus, left hemisphere seizures may propagate more diffusely, causing damage to larger brain regions [48]. In addition, FTBTCS is reportedly more common in LTLE [49]. In the present study, the history of FTBTCS showed bilateral brain damage only in LTLE-HS patients, which supports the previous theory.

### 4.4. Relationship between Abdominal FAS and Subfield Volume Change of AMG in LTLE-HS

FAS frequency was positively correlated with ipsilateral AMG (Me) in RTLE-HS patients, and the ipsilateral AMG volume was less decreased in LTLE-HS patients with abdominal FAS than in those without abdominal FAS.

Although gray matter volume reductions in HC and AMG in the presence of FAS have been reported, the relationship between FAS and the subfield volumes of HC and AMG has been investigated in only a few studies. However, the presence of FAS and surgical outcomes was examined in several previous studies. The presence of FAS is known to be associated with a good surgical outcome owing to the localizing value of FAS [50]. Abdominal FAS has a high localizing value; thus, patients who report experiencing abdominal FAS have a good postoperative prognosis. Conversely, patients who do not experience FAS, which may originate from bitemporal dysfunction or a widespread epileptogenic zone, reportedly have a poor postoperative prognosis [51]. In our study, the volume reduction was less in patients with FAS, which is in agreement with previous findings.

In another study, abdominal FAS did not lateralize the lesions; however, abdominal FAS with “rising sensation” appeared more specific to mesial temporal sclerosis, and an initially epigastric location is more suggestive of left-sided seizure onset [52]. In the present study, abdominal FAS was only associated with subfield volume of AMG among LTLE-HS patients.

Therefore, our results (FAS frequency being positively correlated with AMG volume in RTLE-HS patients and significant AMG volume loss in LTLE-HS patients without abdominal FAS) may be used to predict the postoperative prognosis and interpret the lateralizing role of a variety of FAS in TLE-HS patients.

### 4.5. Limitations of the Present Study

Our study had several limitations. First, the visual confirmation of parcellation in the process of detailed small segmentations using FreeSurfer may have caused limitations. The latest version, FreeSurfer version 6.0, increased the detection of HS by improving the hippocampal subfield segmentation skill to be as similar to the histological subfield as possible [18,53]. However, Peixoto-Santos et al. reported that FreeSurfer version 6.0 could not differentiate HS type 1 from HS type 2, contrary to their manual segmentation on 4.7T MRI [54]. Although FreeSurfer detects hippocampal volume loss with good accuracy [17] and is a useful tool for performing detailed segmentation, the hippocampal subfield analysis should be interpreted with care because it lacks the ability to discriminate between HS types.

Second, since no microscopic approach to actual surgical specimens was performed in this study, it was not possible to determine how accurately subfield analysis using FreeSurfer could predict actual histological subfield segmentation.

Finally, more detailed research is needed regarding the meaning of the subregions showing differences based on laterality. Although the subregions showing changes in the LTLE-HS and RTLE-HS were proven different in the present study, why a difference in that specific subregion was observed could not be explained. Therefore, a larger number of patients should be evaluated in the future to confirm our observations.

In summary, unilateral TLE-HS is a bilateral disease that shows different laterality-dependent characteristics based on the subfield analysis of HC and AMG. Subfield volumes of HC and AMG were associated with clinical variables, and the more damaged substructures depended on laterality in TLE-HS. These findings support the evidence that LTLE-HS and RTLE-HS are disparate epilepsy entities rather than simply identical syndromes harboring a mesial temporal lesion. In addition, the presence of FAS supports good localization value, and abdominal FAS has a high localization value, especially in patients with LTLE-HS.

## Figures and Tables

**Figure 1 medicina-58-00480-f001:**
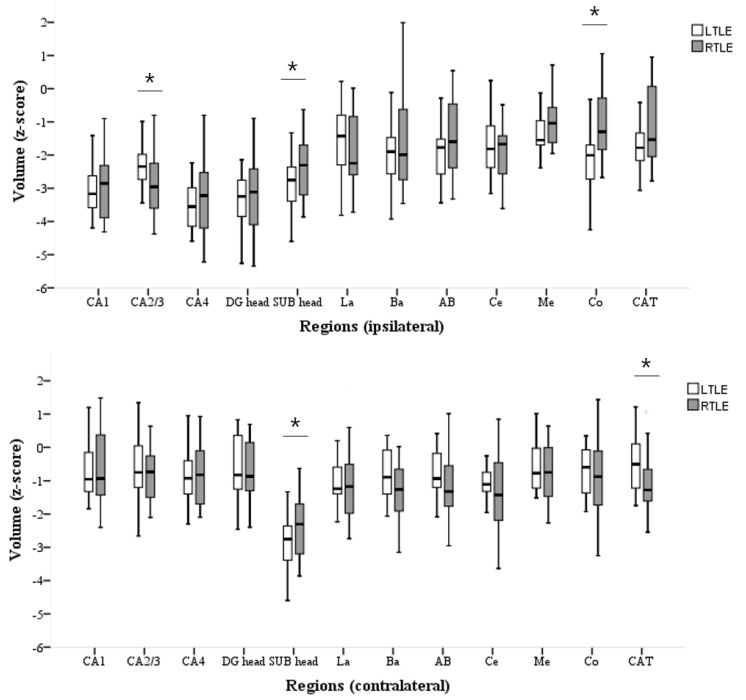
Decreased contralateral hippocampal volumes in patients with left TLE compared to ptients with right TLE (ipsilateral subiculum, contralateral subiculum, ipsilateral cortical nucleus of amygdala) and vice versa (ipsilateral CA2/3, contralateral Cortico-amygdaloid Transition Area). LTLE, left temporal lobe epilepsy with hippocampal sclerosis; RTLE, right temporal lobe epilepsy with hippocampal sclerosis; CA, cornu ammonis; DG, granule cell layer of dentate gyrus; SUB, subiculum; La, Lateral nucleus; Ba, Basal nucleus; AB, Accessory Basal; Ce, Central; Me, Medial; Co, Cortical; CAT, Cortico-amygdaloid Transition Area. * *p*-value < 0.05.

**Figure 2 medicina-58-00480-f002:**
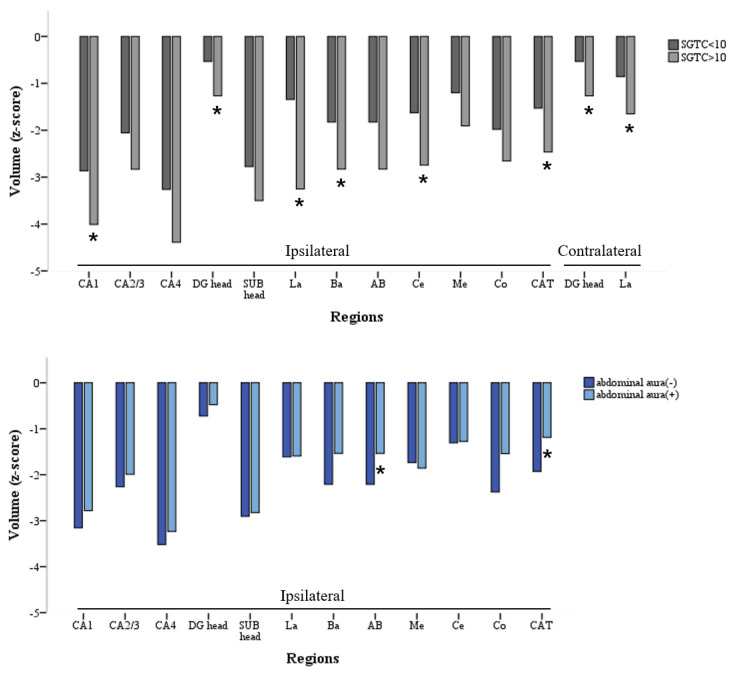
Subfield volume alterations associated with history of more than 10 SGTCS and abdominal aura in left temporal lobe epilepsy with hippocampal sclerosis. CA, cornu ammonis; DG, granule cell layer of dentate gyrus; SUB, subiculum; La, Lateral nucleus; Ba, Basal nucleus; AB, Accessory Basal; Ce, Central; Me, Medial; Co, Cortical; CAT, Cortico-amygdaloid Transition Area. * *p*-value < 0.05.

**Table 1 medicina-58-00480-t001:** Patient characteristics.

	LTLE (*n* = 22)	RTLE (*n* = 26)	HC (*n* = 28)	*p*-Value
Age at MRI (years)	28.7 ± 10.6	34.2 ± 12.1	38.4 ± 6.4	0.084 ^a^
Male, *n* (%)	5 (22.7%)	10 (38.5%)	12 (42.9%)	0.293 ^b^
Age at epilepsy onset (years)	14.8 ± 10.4	17.4 ± 12.8		0.576 ^c^
Duration of epilepsy (years)	16.7 ± 10.4	16.7 ± 12.9		0.780 ^c^
Frequency of FAS (/month)	4.4 ± 5.3	12.2 ± 35.0		0.552 ^c^
Frequency of FIAS (/month)	7.8 ± 18.6	10.2 ± 29.2		0.362 ^c^
History of febrile seizure, *n* (%)	10 (45.5%)	13 (50.0%)		0.753 ^b^
SGTCS > 10, *n* (%)	3 (13.6%)	4 (15.4%)		0.864 ^b^
Abdominal FAS, *n* (%)	8 (36.4%)	18 (69.2%)		0.023 ^b^
Psychic FAS, *n* (%)	12 (54.5%)	7 (26.9%)		0.051 ^b^
Surgical resection, *n* (%)	17 (77.3%)	20 (76.9%)		

LTLE, left temporal lobe epilepsy with hippocampal sclerosis; RTLE, right temporal lobe epilepsy with hippocampal sclerosis; HC, healthy control; FAS, focal aware seizures; FIAS, frequency of focal impaired awareness seizures; SGTCS, secondary generalized tonic-clonic seizures, ^a^ analyzed by ANOVA, ^b^ analyzed by chi-square test, ^c^ analyzed by Mann–Whitney test.

**Table 2 medicina-58-00480-t002:** Correlation between subfield volumes and clinical variables.

			Age at Epilepsy Onset		Duration of Epilepsy		Frequency of FAS		Frequency of FIAS	
Side	Region		LTLE	RTLE	LTLE	RTLE	LTLE	RTLE	LTLE	RTLE
Ipsilateral	Hippocampus	CA1 head	0.204	0.066	−0.499 *	−0.453 *	0.207	0.290	0.151	0.367
		CA1	0.246	0.127	−0.484 *	−0.442 *	0.165	0.291	0.120	0.374
		CA2/3 head	0.079	−0.021	−0.438	−0.421 *	0.310	0.477	0.254	0.261
		CA2/3	0.046	0.101	−0.385	−0.468 *	0.209	0.343	0.154	0.329
		CA4 head	0.144	0.017	−0.600 *	−0.485 *	0.142	0.396	0.046	0.323
		CA4	0.380	0.269	−0.316	−0.475 *	0.078	0.238	−0.148	0.332
		DG head	0.150	0.030	−0.587 *	−0.464 *	0.217	0.385	0.153	0.309
		DG body	0.348	0.280	−0.360	−0.506 *	0.111	0.263	−0.171	0.318
		SUB head	0.419	0.159	−0.165	−0.423 *	−0.091	0.202	−0.076	0.238
		SUB body	0.164	0.358	−0.143	−0.592 *	0.031	0.196	−0.132	0.305
	Amygdala	La	−0.055	0.107	−0.185	−0.388	−0.234	0.072	−0.114	0.082
		Ba	−0.020	0.107	−0.250	−0.375	−0.055	0.108	0.042	0.285
		AB	0.020	0.065	−0.189	−0.313	−0.054	0.114	0.164	0.317
		Ce	0.442 *	0.298	−0.535 *	−0.310	−0.378	−0.020	−0.218	0.322
		Me	0.483 *	0.381	−0.535 *	−0.392 *	−0.074	0.476 *	0.112	0.264
		Co	−0.001	0.306	−0.006	−0.407 *	0.106	0.195	0.210	0.276
		CAT	−0.035	0.024	−0.155	−0.273	−0.045	0.210	0.123	0.327
Contralateral	Hippocampus	CA1 head	−0.123	0.197	−0.287	−0.031	0.292	0.064	−0.015	0.001
		CA1	−0.233	0.280	−0.205	−0.121	0.270	0.029	0.088	−0.014
		CA2/3 head	−0.417	0.256	−0.173	−0.192	0.211	0.036	0.124	−0.080
		CA2/3	−0.445	0.305	−0.097	−0.128	0.226	−0.033	0.124	−0.183
		CA4 head	−0.353	0.239	−0.282	−0.222	0.226	0.096	0.002	−0.005
		CA4	−0.263	0.221	−0.022	−0.196	0.254	0.020	−0.007	−0.328
		DG head	−0.266	0.162	−0.331	−0.135	0.289	0.053	0.052	−0.026
		DG body	−0.199	0.148	−0.027	−0.188	0.221	0.083	−0.050	−0.280
		SUB head	−0.059	0.146	0.035	0.000	0.015	0.096	0.008	0.054
		SUB body	0.052	0.272	0.122	−0.273	0.177	−0.235	−0.113	−0.136
	Amygdala	La	−0.046	0.300	−0.323	−0.234	0.071	0.264	−0.123	0.014
		Ba	−0.177	0.196	−0.322	−0.120	0.227	0.213	0.020	0.108
		AB	−0.187	0.208	−0.278	−0.020	0.381	0.057	0.239	−0.038
		Ce	0.069	0.116	−0.306	−0.048	0.052	0.125	−0.007	0.126
		Me	0.153	0.074	−0.406	0.027	−0.092	0.225	0.285	0.320
		Co	−0.073	0.234	−0.301	−0.021	0.221	0.060	0.342	−0.076
		CAT	−0.341	0.183	−0.170	0.092	0.364	0.125	0.221	0.013

In LTLE, age at epilepsy onset correlated with the volumes of substructures of ipsilateral AMG (Ce and Me), and epilepsy duration correlated with volumes of substructures of ipsilateral HC (CA1 head, CA1, CA4 head, and DG head) and AMG (Ce and Me). In RTLE, epilepsy duration was related to the volumes of substructures of ipsilateral HC (CA1 head, CA1, CA2/3 head, CA2/3, CA4 head, CA4, DG head, DG body, SUB head and SUB body) and AMG (Me and Co), and frequency of FAS was related to volumes of substructures of ipsilateral AMG (Me). Spearman rho values are presented, * *p*-value < 0.05, LTLE, left temporal lobe epilepsy with hippocampal sclerosis; RTLE, right temporal lobe epilepsy with hippocampal sclerosis; FAS, focal aware seizures; FIAS, focal impaired awareness seizures; CA, cornu ammonis; DG, granule cell layer of dentate gyrus; SUB, subiculum; La, Lateral nucleus; Ba, Basal nucleus; AB, Accessory Basal; Ce, Central; Me, Medial; Co, Cortical; CAT, Cortico-amygdaloid Transition Area.

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
