# Peer review of "Lateralizing Characteristics of Morphometric Changes to Hippocampus and Amygdala in Unilateral Temporal Lobe Epilepsy with Hippocampal Sclerosis"

_medicina, 2022, doi:10.3390/medicina58040480_

Round 1

Reviewer 1 Report

In the present study, Dr. Jo and colleagues sought to investigate the substructural volume change in the hippocampus and amygdala in patients with mesial temporal lobe epilepsy with hippocampal sclerosis. They found that the patients with unilateral temporal lobe epilepsy with hippocampal sclerosis is bilateral disease which shows different laterality-dependent characteristics based on the subfield analysis of hippocampus and amygdala. Exclusion and inclusion criteria of patients selection can be added in method participants. This study provides important findings that LTLE-HS and RTLE-HS are different epilepsy entities. 

Reviewer 2 Report

 thanks to the Authors who perfectly designed and did this study that lateralized the morphology of amygdal and hippocampus in TLE.

I suggest you should add:

1- Some MR imaging Data

2-Shorter the abstract

3-slightly stress on the clinical utility of the findings in the discussion .

Additional comments:

1-The main question addressed by the research is the lateralizing lesions in specific areas of the brain.It confirms the previous works
2-Paper address a specific gap in the field and confirm the bilaterality of the damage.
3-The previous studies may not address the bilaterality of the damage.
3-Methodology is very good and the researchers did it well.
4-I assume the conclusion is proper
However:
the abstract should be revised
2- Some tables should be less complicated or should be better explained by the sub titles.

Best regards.
